# Quality of reported ages: A robust re-modification in Total Modified Whipple's Index

**Afza Rasul** [1,2,3]ʘ*, **Jamal Abdul Nasir** [2]ʘ, **Domantas Jasilionis**[3‡], **Dmitri A. Jdanov**[3,4‡]

**1** Department of Statistics, Lahore College for Women University, Lahore, Punjab, Pakistan, **2** Department of Statistics, Government College University, Lahore, Punjab, Pakistan, **3** Laboratory of Demographic Data, Max Planck Institute for Demographic Research, Rostock, Mecklenburg, Germany, **4** National Research University Higher School of Economics, Moscow, Russia

ʘ These authors contributed equally to this work.
‡ These authors contributed equally to this work.
* afza.rasul@lcwu.edu.pk

## Abstract

In demographic research, the accuracy of the reported ages in surveys and censuses is a persistently important issue. The common indices developed and used to examine the quality of age data are Whipple, Myers, Bachi's, modified Whipple, and the total modified Whipple's index. The most commonly used and simplest to compute index is the original Whipple's index proposed by George Chandler Whipple. It is a summary measure used to check age heaping on ages ending with digits 0 and 5. The other summary index is the total modified Whipple index by Spoorenberg. A robust modification is proposed for the total modified Whipple index. This modification, based on the method of the original Whipple index for all digits (0, 1, 2 … 9), is simple, robust, and easy to interpret. The proposed index is used to check the quality of age data from the Demographic and Health Survey data series of three countries (India, Pakistan, and Turkey). The original Whipple index only measures the preferences or avoidances of ages ending in digits 0 and 5. The existing summary index total modified Whipple index, based on modified Whipple indexes, covers all ages; however, results showed that the digit indexes for the terminal digits 1, 2, 3 and 4 have some added probability of being included. The newly proposed summary index, Robust Modified Whipple Index, is based on Rogers Whipple indexes, which display an equal probability for all ten digits $i = 1,2,3…,9$. In the application to the series of datasets, the proposed index is found better and more robust. The comparison of the results of the total modified Whipple index and proposed robust modified Whipple index concludes that the proposed modification is a more precise and robust to measure age misreporting by taking the effects of all terminal digits of reported ages. The proposed modification is suitable for the evaluation of the quality of self reported and interviewer recorded ages at the time of surveys and censuses.

**Data availability statement:** The data used in this study were obtained from the Demographic and Health Surveys (DHS) Program. These datasets are publicly available but require registration and approval. The study utilized all available waves of DHS data for India, Pakistan, and Turkey. Interested researchers can request access by creating an account and submitting a data request through the DHS Program website (https://dhsprogram.com/data/dataset_admin/login_main.cfm). Upon approval, the relevant datasets can be accessed from the DHS Program (https://dhsprogram.com/data/available-datasets.cfm).

**Funding:** The author(s) received no specific funding for this work.

**Competing interests:** The authors have declared that no competing interests exist.

## Introduction

Age is one of the most important study variables in demographic research. Accurate information on respondents' ages and the population's age structure is critically important. Misreporting of age can lead to incorrect statistical estimates and may mislead policymakers when formulating effective policies. Therefore, understanding the quality of age data is important before conducting any analysis.

Age statements are commonly affected by two types of errors: age heaping, which is the tendency to round ages to specific digits (usually 0 or 5), and systematic exaggeration or underestimation of age. Age heaping, also known as digit preference, often occurs when individuals do not know their exact age and either report a rounded number or ask enumerators to record an approximate age they consider suitable. Age heaping and misreporting are critical issues in many developing countries [1]. Researchers' findings have shown that age heaping and digit preference are present in many Demographic and Health Surveys (DHS) [2–5]. Similarly, age misreporting is commonly observed in other surveys and national censuses across developing countries [4, 6–10], often leading to distortions in age distributions. Understanding patterns in reported ages is essential for assessing the quality of age data.

The debate about age misreporting among demographers is not a new one. A century ago, an American demographer, George Chandler Whipple (1866–1924), gave an index to measure the tendency for human age misreporting. Whipple introduces the index as a ratio of the number of people reporting their age as a multiple of 5 by one-fifth of the total sample/population size in the age range of 23–62 years [11]. It assumes a linear distribution of ages in a 5-year age range. Younger ages (0–22) and older ages (63 and above) are excluded as the linearity assumption is not plausible for these ages. Later, other techniques [12,13] have been developed and used to measure age misreporting in surveys and censuses for age distributions. However, the Whipple index remains the most well-known among all other indices due to its underlying properties and simple calculation.

Demographers have also been trying to introduce new indices to measure the accuracy of age reporting. Whipple index follows the assumption of rectangular distribution, which assumes an equal number of persons in each 10-year age group. It identifies the preference (avoidance) of terminal digits of an age ending at '0' and '5'. Whipple used the range 23–62 (both inclusive) years of age for the calculation of the quality index for age data. This age range can be changed, but the rectangular distribution and linearity assumptions must be achieved. The attractive benefit of the Whipple index is that it is valid for a single year of age distribution. However, the Whipple index only measures age heaping at two digits, multiples of 5 (ending with 0 or 5), and fails to express age misreporting at the other 8 digits.

## From original Whipple to modified versions

Original Whipple's index (*WI*) is calculated as a ratio of the people reporting their ages as a multiple of 5 (terminal digit 0 or 5) and the persons in the age range 23 and 62.

$$WI = \{5\left(P_{25} + P_{30} + P_{35} + \ldots + P_{55} + P_{60}\right) / \left(P_{23} + P_{24} + P_{25} + \ldots + P_{61} + P_{62}\right)\} * 100 \tag{1}$$

Where $P_x$ is the population reporting age $x$ in completed years. It is based on the rectangular distribution property and assumes a linear distribution of ages in a 5-year age range. In this original version, the Whipple index only checks the preferences (avoidances) at the ages with terminal digits of 0 and 5 without any distinctions. To cover this, the first change was suggested by Roger, observed in the book [14], which distinguishes the preferences for ages ending with terminal digits 0 and 5.

$$WI_0 = 10\left(P_{30} + P_{40} + P_{50} + P_{60}\right) / \left(P_{23} + P_{24} + P_{25} + \ldots + P_{62}\right) \tag{2}$$

$$WI_5 = 10\left(P_{25} + P_{35} + P_{45} + P_{55}\right) / \left(P_{23} + P_{24} + P_{25} + \ldots + P_{62}\right) \tag{3}$$

[For ease of presentation multiplier 100 is omitted from equation (2) to onward]. By taking the arithmetic mean of equations (2) and (3), we can get equation (1), i.e., original Whipple Index (*WI*)

$$WI = \frac{WI_0 + WI_5}{2} \tag{4}$$

This modification provides a meaning to distinguish between the ages ending with terminal digits 0 and 5, based on the property of rectangular distribution and linearity assumption over 10 years, like the original *WI*.

The second modification was suggested by Noumbissi [15], is based on the more sensible assumption of linearity across five years rather than a 10-year age range. Noumbissi feels that the 10-year assumption in the formulation of Roger's and the original Whipple index is crude. Therefore, he suggested a modification in the Roger's Whipple index, based on a more reasonable assumption of linearity, taking the age range of 5 years for terminal digits '0' and '5' [15]. He suggested the following formulas to calculate heaping at terminal digits '0' and '5':

$$W_0 = 5\left(P_{30} + P_{40} + P_{50} + P_{60}\right) / \left({}_5P_{28} + {}_5P_{38} + {}_5P_{48} + {}_5P_{58}\right) \tag{5}$$

$$W_5 = 5\left(P_{25} + P_{35} + P_{45} + P_{55}\right) / \left({}_5P_{23} + {}_5P_{33} + {}_5P_{43} + {}_5P_{53}\right) \tag{6}$$

Here, $P_x$ is the number of individuals reporting their age as $x$ years, and ${}_5P_x$ is the number of individuals reporting their age in the age range *(x, x+4)*. Similarly, age heaping can be calculated for other terminal digits *(1,2, …, 9) as:*

$$W_1 = 5\left(P_{31} + P_{41} + P_{51} + P_{61}\right) / \left({}_5P_{29} + {}_5P_{39} + {}_5P_{49} + {}_5P_{59}\right) \tag{7}$$

$$W_2 = 5\left(P_{32} + P_{42} + P_{52} + P_{62}\right) / \left({}_5P_{30} + {}_5P_{40} + {}_5P_{50} + {}_5P_{60}\right) \tag{8}$$

$$W_3 = 5\left(P_{23} + P_{33} + P_{43} + P_{53}\right) / \left({}_5P_{21} + {}_5P_{31} + {}_5P_{41} + {}_5P_{51}\right) \tag{9}$$

$$W_4 = 5\left(P_{24} + P_{34} + P_{44} + P_{54}\right) / \left({}_5P_{22} + {}_5P_{32} + {}_5P_{42} + {}_5P_{52}\right) \tag{10}$$

$$W_6 = 5\left(P_{26} + P_{36} + P_{46} + P_{56}\right) / \left({}_5P_{24} + {}_5P_{34} + {}_5P_{44} + {}_5P_{54}\right) \tag{11}$$

$$W_7 = 5\left(P_{27} + P_{37} + P_{47} + P_{57}\right) / \left({}_5P_{25} + {}_5P_{35} + {}_5P_{45} + {}_5P_{55}\right) \tag{12}$$

$$W_8 = 5\left(P_{28} + P_{38} + P_{48} + P_{58}\right) / \left({}_5P_{26} + {}_5P_{36} + {}_5P_{46} + {}_5P_{56}\right) \tag{13}$$

$$W_9 = 5\left(P_{29} + P_{39} + P_{49} + P_{59}\right) / \left({}_5P_{27} + {}_5P_{37} + {}_5P_{47} + {}_5P_{57}\right) \tag{14}$$

When there is no digit preference/avoidance, these digit-specific modified Whipple indices are equal to 1 [15].

The third modification is suggested by Spoorenberg [16]. He felt the extensions proposed by Noumbissi to all ten digits are not practical for spatial, temporal, and spatiotemporal comparisons. He introduced a summary index based on the modified Whipple index proposed by Noumbissi and named it the Total Modified Whipple Index ($W_{tot}$).

$$W_{tot} = \sum_{i=0}^{9} \left(|W_i - 1|\right) \tag{15}$$

where $W_i$ is the digit specified modified Whipple index for each of the ten digits (0–9) developed by Noumbissi.

**If no preference is observed, then**

$W_0 = W_1 = W_2 = W_3 = W_4 = W_5 = W_6 = W_7 = W_8 = W_9 = 1$ and

$$W_{tot} = \sum_{i=0}^{9} \left(|W_i - 1|\right) = 0$$

If all reported ages end in 0 or 5, then $W_0 = W_5 = 5$ and all other $W_i = 0$. Hence, $W_{tot}$ reaches the maximum value of 16. This index can be used as a general measure of the quality of age reporting in complement to Noumbissi's previous development [16].

**Mathematical inconsistencies in Noumbissi's Indexes**

There are a few shortcomings in the calculation of the equation for individual digits given in equations (5–14) by Noumbissi. If we expand the numerator of the above equations (5–14), the equations will be as follows;

$$\begin{aligned}W_0 = 5\left(P_{30} + P_{40} + P_{50} + P_{60}\right) / \{&\left(P_{28} + P_{29} + P_{30} + P_{31} + P_{32}\right) + \left(P_{38} + P_{39} + P_{40} + P_{41} + P_{42}\right) \\ &+ \left(P_{48} + P_{49} + P_{50} + P_{51} + P_{52}\right) + \left(P_{58} + P_{59} + P_{60} + P_{61} + P_{62}\right)\}\end{aligned} \tag{5e}$$

$$\begin{aligned}W_5 = 5\left(P_{25} + P_{35} + P_{45} + P_{55}\right) / \{&\left(P_{23} + P_{24} + P_{25} + P_{26} + P_{27}\right) + \left(P_{33} + P_{34} + P_{35} + P_{36} + P_{37}\right) \\ &+ \left(P_{43} + P_{44} + P_{45} + P_{46} + P_{47}\right) + \left(P_{53} + P_{54} + P_{55} + P_{56} + P_{57}\right)\}\end{aligned} \tag{6e}$$

$$\begin{aligned}W_1 = 5\left(P_{31} + P_{41} + P_{51} + P_{61}\right) / \{&\left(P_{29} + P_{30} + P_{31} + P_{32} + P_{33}\right) + \left(P_{39} + P_{40} + P_{41} + P_{42} + P_{43}\right) \\ &+ \left(P_{49} + P_{50} + P_{51} + P_{52} + P_{53}\right) + \left(P_{59} + P_{60} + P_{61} + P_{62} + P_{63}\right)\}\end{aligned} \tag{7e}$$

$$W_2 = 5\left(P_{32} + P_{42} + P_{52} + P_{62}\right) / \{(P_{30} + P_{31} + P_{32} + P_{33} + P_{34}) + (P_{40} + P_{41} + P_{42} + P_{43} + P_{44})$$
$$+ (P_{50} + P_{51} + P_{52} + P_{53} + P_{54}) + (P_{60} + P_{61} + P_{62} + P_{63} + P_{64})\}$$

(8e)

$$W_3 = 5\left(P_{23} + P_{33} + P_{43} + P_{53}\right) / \{(P_{21} + P_{22} + P_{23} + P_{24} + P_{25}) + (P_{31} + P_{32} + P_{33} + P_{34} + P_{35})$$
$$+ (P_{41} + P_{42} + P_{43} + P_{44} + P_{45}) + (P_{51} + P_{52} + P_{53} + P_{54} + P_{55})\}$$

(9e)

$$W_4 = 5\left(P_{24} + P_{34} + P_{44} + P_{54}\right) / \{(P_{22} + P_{23} + P_{24} + P_{25} + P_{26}) + (P_{32} + P_{33} + P_{34} + P_{35} + P_{36})$$
$$+ (P_{42} + P_{43} + P_{44} + P_{45} + P_{46}) + (P_{52} + P_{53} + P_{54} + P_{55} + P_{56})\}$$

(10e)

$$W_6 = 5\left(P_{26} + P_{36} + P_{46} + P_{56}\right) / \{(P_{24} + P_{25} + P_{26} + P_{27} + P_{28}) + (P_{34} + P_{35} + P_{36} + P_{37} + P_{38})$$
$$+ (P_{44} + P_{45} + P_{46} + P_{47} + P_{48}) + (P_{54} + P_{55} + P_{56} + P_{57} + P_{58})\}$$

(11e)

$$W_7 = 5\left(P_{27} + P_{37} + P_{47} + P_{57}\right) / \{(P_{25} + P_{26} + P_{27} + P_{28} + P_{29}) + (P_{35} + P_{36} + P_{37} + P_{38} + P_{39})$$
$$+ (P_{45} + P_{46} + P_{47} + P_{48} + P_{49}) + (P_{55} + P_{56} + P_{57} + P_{58} + P_{59})\}$$

(12e)

$$W_8 = 5\left(P_{28} + P_{38} + P_{48} + P_{58}\right) / \{(P_{26} + P_{27} + P_{28} + P_{29} + P_{30}) + (P_{36} + P_{37} + P_{38} + P_{39} + P_{40})$$
$$+ (P_{46} + P_{47} + P_{48} + P_{49} + P_{50}) + (P_{56} + P_{57} + P_{58} + P_{59} + P_{60})\}$$

(13e)

$$W_9 = 5\left(P_{29} + P_{39} + P_{49} + P_{59}\right) / \{(P_{27} + P_{28} + P_{29} + P_{30} + P_{31}) + (P_{37} + P_{38} + P_{39} + P_{40} + P_{41})$$
$$+ (P_{47} + P_{48} + P_{49} + P_{50} + P_{51}) + (P_{57} + P_{58} + P_{59} + P_{60} + P_{61})\}$$

(14e)

It can be seen that all expanded equations (5e–14e) are not based on a complete age range (23–62) years. Moreover, for equations (7e, 8e, 9e, and 10e), few values of the denominator are beyond the age range (23–62). These equations give unrealistic and poor predictions of the indexes for digits (1, 2, 3, and 4). Therefore, there is an obvious shortcoming in the indexes proposed by Noumbissi. The approach to calculating individual digit (0–9) preferences in Noumbissi's version is unrealistic and not based on mathematical formulation. Spoorenberg's total modified Whipple index ($W_{tot}$) is based on Noumbissi's modification. Therefore, it does not give an appropriate summary value of all terminal digits robustly.

## Theoretical foundation

The evolution of the Whipple index from its original form to the modified versions reflects a continuous effort in demography to refine statistical tools to measure age misreporting. However, the existing methods for quantifying age misreporting, such as the modified Whipple's Index proposed by Noumbissi and subsequently adopted by Spoorenberg, exhibit limitations in their mathematical formulation, particularly when calculating indices for individual terminal digits. An examination of the expanded equations for individual digit indices (equations 5e-14e from Noumbissi's work) reveals that the denominators do not consistently encompass the entire age range of interest (typically 23–62 years). Furthermore, for several digits (specifically 1, 2, 3, and 4, corresponding to equations 7e, 8e, 9e, and 10e), the denominator includes age values falling outside this defined range. This inconsistency in the reference age range leads to potentially unrealistic and

inaccurate predictions of the individual digit indices. Consequently, the approach used by Noumbissi for calculating individual digit preferences appears mathematically flawed and lacks a robust theoretical basis.

Since Spoorenberg's total modified Whipple index ($W_{tot}$) is derived from Noumbissi's modification, it inherits these shortcomings and may not provide a reliable summary measure of age misreporting across all terminal digits. A need exists for a summary index that accurately captures age misreporting or heaping across the entire specified data range (23–62 years) without the aforementioned mathematical inconsistencies. To address these limitations of existing indices, this research proposes a re-modification of the Total Modified Whipple Index. The new index is constructed in the logic of the original Whipple index, applying its structure systematically to each terminal digit using the mathematically consistent framework of the Rogers Whipple Indices ($WI_i$). In doing so, this research aims to contribute to the ongoing development of demographic quality measures by presenting the Robust Modified Whipple Index ($RMWI$) as a more reliable and comprehensive tool for assessing age misreporting.

## Methods

This section introduces a new measure to assess the quality of age data: the Robust Modified Whipple Index ($RMWI$). In the next section, we also proposed a criterion for interpreting $RMWI$ values to evaluate the accuracy of reported age data.

### Robust Modified Whipple Index ($RMWI$) [Proposed Index]

Following the Roger's Whipple index, the indexes for each terminal digit (0–9) can be calculated as;

$$WI_0 = 10 \left(P_{30} + P_{40} + P_{50} + P_{60}\right) / \left(P_{23} + P_{24} + P_{25} + \ldots + P_{62}\right) \tag{15}$$

$$WI_1 = 10 \left(P_{31} + P_{41} + P_{51} + P_{61}\right) / \left(P_{23} + P_{24} + P_{25} + \ldots + P_{62}\right) \tag{16}$$

$$WI_2 = 10 \left(P_{32} + P_{42} + P_{52} + P_{62}\right) / \left(P_{23} + P_{24} + P_{25} + \ldots + P_{62}\right) \tag{17}$$

$$WI_3 = 10 \left(P_{23} + P_{33} + P_{43} + P_{53}\right) / \left(P_{23} + P_{24} + P_{25} + \ldots + P_{62}\right) \tag{18}$$

$$WI_4 = 10 \left(P_{24} + P_{34} + P_{44} + P_{54}\right) / \left(P_{23} + P_{24} + P_{25} + \ldots + P_{62}\right) \tag{19}$$

$$WI_5 = 10 \left(P_{25} + P_{35} + P_{45} + P_{55}\right) / \left(P_{23} + P_{24} + P_{25} + \ldots + P_{62}\right) \tag{20}$$

$$WI_6 = 10 \left(P_{26} + P_{36} + P_{46} + P_{56}\right) / \left(P_{23} + P_{24} + P_{25} + \ldots + P_{62}\right) \tag{21}$$

$$WI_7 = 10 \left(P_{27} + P_{37} + P_{47} + P_{57}\right) / \left(P_{23} + P_{24} + P_{25} + \ldots + P_{62}\right) \tag{22}$$

$$WI_8 = 10 \left(P_{28} + P_{38} + P_{48} + P_{58}\right) / \left(P_{23} + P_{24} + P_{25} + \ldots + P_{62}\right) \tag{23}$$

$$WI_9 = 10 \left(P_{29} + P_{39} + P_{49} + P_{59}\right) / \left(P_{23} + P_{24} + P_{25} + \ldots + P_{62}\right) \tag{24}$$

Based on indexes ($WI_0$ to $WI_9$) in the equations (15–24), a modified version of Spoorenberg's total modified Whipple index ($W_{tot}$) can be calculated as;

$$RMWI = \sum_{i=0}^{9} \left( \left| WI_i - 1 \right| \right)$$

(25)

**If there is no heaping at any digits, then**

$WI_0 = WI_1 = WI_2 = WI_3 = WI_4 = WI_5 = WI_6 = WI_7 = WI_8 = WI_9 = 1$ and

$$RMWI = \sum_{i=0}^{9} \left( |1 - 1| \right) = 0$$

If all reported ages are heaped at a multiple of 10, i.e., end at digit 0, then $WI_0 = 10$ and all other $WI_i = 0$, and hence, $RMWI = 18$.

The analysis in this study focuses on the age range of 23–62 years, a commonly used span in demographic research that helps reduce distortions from age misreporting, especially at younger and older ages. This range effectively captures the core adult population, where age heaping is typically most evident, making it a suitable base for evaluating digit preference or avoidance. However, the proposed Robust Modified Whipple Index ($RMWI$) is flexible and can be applied to any rectangular age distribution where each terminal digit (0–9) has an equal probability. For example, alternative ranges such as 15–64 or 20–79 can be used depending on the research objectives and the quality of the available data.

**A proposed criterion for the quality of age reporting for *RMWI***

To smooth the interpretation of the *RMWI* value and provide a standardized assessment of age reporting quality, a criterion is proposed that parallels the widely used United Nations (UN) standards for the original Whipple Index. The United Nations proposed the following (Table 1) criteria to assess the quality of age reporting based on the Whipple Index [17].

The original Whipple's index is a summary measure of the quality of age data, which emphasizes the heaping at digits 0 and 5. It ignores the digit preferences (avoidance) on all other 8 digits (1, 2, 3, 4, 6, 7, 8, and 9). The *RMWI*, by incorporating the deviations from expectation for all ten terminal digits, offers a more comprehensive measure of age misreporting. A set of ranges for the *RMWI* value is established to categorise the quality of age reporting (Table 2). To calculate the range for *RMWI*, the following cases are considered

1. If all ages are correctly reported, i.e., $WI_0 = WI_1 = WI_2 = \ldots = WI_9 = 1$, $RMWI = 0$.

2. If all reported ages are heaped at a multiple of 10, i.e., end at digit 0, $WI_0 = 10$ and all other $WI_i = 0$, $RMWI = 18$. Similarly, if all reported ages ended in a single digit, i.e., any one $WI_i = 10$ and all other $WI_i = 0$, $RMWI = 18$.

3. If any two digits are reported only (say 0 and 5), then $WI_0 = WI_1 = 5$ and all other $WI_i = 0$, $RMWI = 16$.

**Table 1. UN criteria for the quality of reported ages.**

| Whipple Index Value | Deviation from perfection | Quality of data |
|---|---|---|
| <105 | <5% | Perfectly Accurate |
| 105–110 | 5–9.99% | fairly Accurate |
| 110–125 | 10–24.99% | Moderate |
| 125–175 | 25–74.99% | Poor/rough |
| >175 | ≥ 75% | Very poor/rough |

**Table 2. Criteria for quality of reported ages using *RMWI*.**

| *RMWI* Value | Quality of data | % of People reporting incorrect Age |
|---|---|---|
| 0.00–0.19 | Perfectly Accurate | <1% |
| 0.20–0.99 | fairly Accurate | 1–4.99% |
| 1.00–1.99 | Moderate | 5–9.99% |
| 2.00–2.99 | Poor/rough | 10–14.99% |
| >3 | Very poor/rough | ≥ 15% |

Hence, the value of *RMWI* ranges between 0 and 18. A value of "0" represents the perfection of data. As the value of *RMWI* deviates from 0, it approaches the imperfection of the reported ages, implying that the ages are misreported. These theoretical bounds of the *RMWI* (ranging from 0 to 18) represent benchmarks for perfect accuracy and extreme misreporting, respectively. Based on the index's structure and its sensitivity to varying patterns of digit preference, the following ranges are proposed to classify the quality of age data based on *RMWI* (Table 2).

The proposed classification in Table 2 offers a practical scale for assessing the quality of age reporting using the *RMWI* index. A *RMWI* value between 0.00 and 0.19 indicates perfect accuracy of age reporting, reflecting negligible digit preference, with less than 1% of individuals misreporting their age. Values from 0.20 to 0.99 suggest fairly accurate data, indicating 1% to 4.99% age misreporting likely due to weak digit preference. A range of 1.00 to 1.99 signals moderate age misreporting (5% to 9.99%), indicating clear evidence of age misreporting. Values between 2.00 and 2.99 represent poor or rough data quality, pointing to widespread age misreporting. Finally, a *RMWI* value exceeding 3.00 suggests very poor data quality, where at least 15% of individuals are likely misreporting their ages. The percentage ranges of *RMWI* are derived from the mathematical structure and its sensitivity to departures from uniformity of terminal digits. A higher *RMWI*

**Table 3. Sample size of the individual respondent in the DHS series of three selected countries.**

| Countries/ years | Total Respondents | Study sample Aged 23–62 |
|---|---|---|
| **India** | | |
| 2005–2006 | 534,161 | 243,520 |
| 2015–2016 | 2,869,043 | 1,380,023 |
| 2019–2021 | 2,843,917 | 1,437,924 |
| **Turkey** | | |
| 1993 | 40,840 | 17,841 |
| 1998 | 37,991 | 16,899 |
| 2003 | 47,894 | 22,161 |
| 2008 | 44,498 | 21,522 |
| 2013 | 45,660 | 22,807 |
| 2018 | 39,914 | 20,308 |
| **Pakistan** | | |
| 1990–1991 | 52,358 | 18,293 |
| 2006–2007 | 727,493 | 264,060 |
| 2012–2013 | 94,169 | 36,647 |
| 2017–2018 | 100,869 | 39,982 |

Data Source: Standard DHS of India (2005–2021), Turkey (1993–2018) and Pakistan (1990–2018) Household data files [18]

reflects a greater deviation from expected digit uniformity, suggesting more significant age misreporting. This interpretive framework adds practical value to the *RMWI* by enabling comparisons across datasets and populations. By incorporating all ten terminal digits with equal weight and offering a robust summary measure, the *RMWI* and its associated criterion provide a more comprehensive and statistically consistent approach than earlier modifications.

The following sections illustrate the practical application of the *RMWI* and this criterion using real-world and simulated datasets, demonstrating its effectiveness in identifying and quantifying age misreporting.

## Results

### Application to data

To test the proposed index, *RMWI*, we applied it to the Standard Demographic and Health Survey (DHS) datasets from three countries: India, Turkey, and Pakistan. A comparison with existing indies was carried out to enhance the analysis. Standard DHS surveys collect primary data using four types of model questionnaires. Among these questionnaires, a household questionnaire is used to collect data on the characteristics of the household for all the usual residents of the household. This is the main questionnaire that includes all respondents who are used to get information for other types of questionnaires. In this household schedule, information is available on age, sex, education, relationship to household head, residence, and other details. Reported age data from individuals from this household questionnaire is used for this research. The sample size used for this study is given in Table 3 [18]. More details about the DHS surveys can be found at the DHS website (www.dhsprogram.com).

Among the three selected countries for analysis, the quality of age reporting is considered poor in India and Pakistan [2], and average in Turkey. Appendix Tables A1, and A2 give the digit-specific Whipple indexes ($WI_i$) and digit-specific modified Whipple indexes ($W_i$) for India, Turkey, and Pakistan respectively.

Fig 1 shows the digit preference (avoidance) using the digit-specific Whipple indexes $WI_i$ (Roger's indexes) and modified Whipple indexes $W_i$ (Noumbissi's indexes). Technically, if all digits are correctly reported, then the value of all digit-specific indexes must be 1. If the value of an index for any digit is greater than 1, it is the preferred digit by people telling their age at the time of data collection. This may cause heaping at those digits. If the value of the index is less than 1, this shows the avoidance of that digit.

Age reporting in India and Pakistan follows a classical pattern of age misreporting. A strong age heaping at the digits 0 and 5 is reflected in both digits' specific Whipple index as well as in the original Whipple index. Digits 1 and 9 are the most avoided in age reporting in both countries. Preferences for 0 and 5 have declined a little bit over time but the situation of the accurate reporting of age is still looking like a nightmare in both countries. The situation in Turkey is less extreme. Preferences for 0 and 5 declined over time and were very near to 1 in the year 2008 to onwards. In Turkish datasets, digit 3 is as likely to be preferred as 0 or 5. The most observed pattern of digit preference in India is 5, 0, and 8 for digit-specific Whipple's index ($WI_i$) and 5, 0, and 2 for digit-specific Modified Whipple's index ($W_i$). The digit "2" replaces its position from digit "8" due to the overestimation of digit "2" in the digit-specific Modified Whipple's index in all three-survey datasets of India. Similarly, digit "3" also changes its position over digit "4" due to the overestimation of the index for digit 3. Although the digits "1 and 4" have not changed their position these two digits also overestimated the digit in the digit-specific Modified Whipple's index. In the case of Pakistan, digit "5" is ranked as the highest preferred digit in digit-specific Whipple indexes; however, for the PDHS 2005–06 and PDHS 2012–13, the digit "0" ranked as the highest preferred digit in the digit-specific modified Whipple index. Similarly, digits 2 and 3 ranked at earlier positions due to overestimation in the digit-specific modified Whipple index. In Turkish datasets, all overestimated digits "1, 2, 3, and 4" in digit-specific modified Whipple indexes moved to earlier ranks as compared to digit-specific Whipple indexes. In the digit-specific modified Whipple index (Noumbissi's indexes), digit 3 is preferred over digit 5 due to the unstable calculating equation for digit 3. Digits 1 and 9 are the most avoided in all populations (see Fig 1, Appendix A1 and A2).

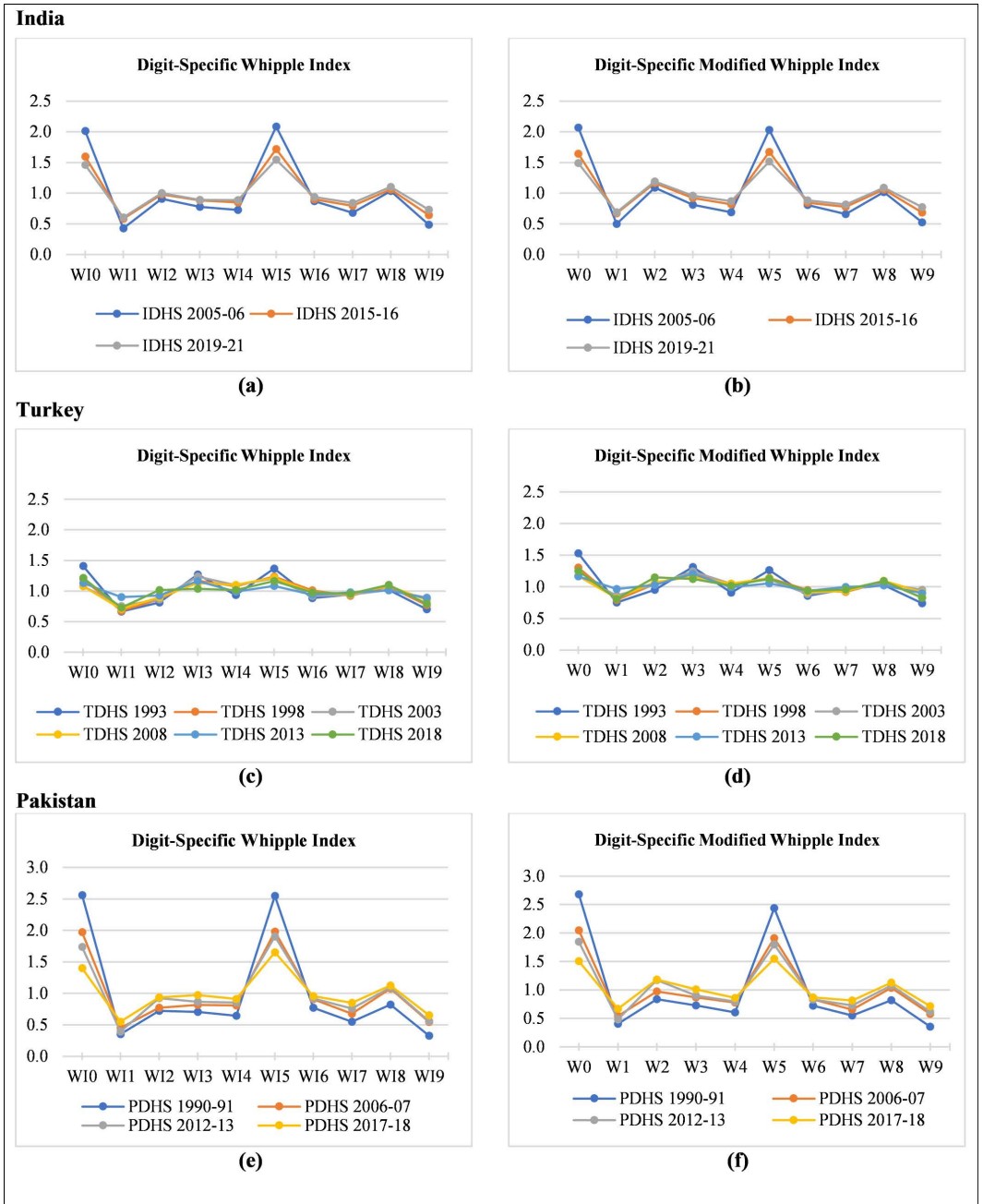

**Fig 1. Quality of age reporting: Digit-specific Whipple's index (*WIi*) and digit-specific Modified Whipple's index (*Wi*) for the series of DHS data of India, Turkey, and Pakistan.** Data Source: Standard DHS of India (2005–2021), Turkey (1993–2018) and Pakistan (1990–2018) [18], Figures based on the Author's calculations of digit-specific Whipple indexes.

## Comparison of digit specific indexes: $W_i$ vs $WI_i$

The accuracy of the total index (calculated for all ages) depends on the accuracy of digit-specific indexes. In this section, we compare two digit-specific indexes used to calculate the total modified Whipple index ($W_{tot}$) and the proposed *RMWI* index.

If no preference (avoidance) of any specific digit is present, then the digit specified index for any single terminal digit must be equal to 1, corollary, sum of all ten indexes must be equal to 10. Statistically, for any population where age is reported correctly, then the probability of each terminal digit of age must be equal to 0.1, and $\sum_{i=0}^{9} P_i = 1$ for all terminal digits i = 0, 1, 2, …, 9. Practically, some digits are preferred ($P_i > 0.1$), and some are avoided ($P_i < 0.1$), however, the sum of all digit probabilities must be 1, and corollary, the sum of all digit indexes (DIs) must be 10. Therefore, in any population following a rectangular distribution and assuming the linearity assumption, the following results can be obtained for individual terminal digit probability ($P_i$) and digit index ($DI_i$).

P_i < 0.1    =>    $DI_i < 1$ Avoided digit (under-reporting of age)

P_i = 0.1    =>    $DI_i = 1$ Ideal digit (correct reporting of age)

P_i > 0.1    =>    $DI_i > 1$ Preferred digit (over-reporting of age)

$\sum_{i=0}^{9} P_i = 1$ or $\sum_{i=0}^{9} DI_i = 10$

Digit-specified indexes calculated in tables Appendix A1 and A2 showed that in all thirteen datasets of three countries, few digits are preferred ($DI_i > 1$) and others are avoided digits ($DI_i < 1$). This is expected in all countries with age misreporting. As a result, the probability of individual terminal digits is greater than 0.1 (preferred) or less than 0.1 (avoided). To compare the age misreporting indices, look deeply at Appendix A1 and A2. In the calculation of digit specified indexes, two methods, W (modified Whipple i.e. Noumbissi's indexes), and WI (original Whipple), are used. Apparently, all digit-specified indexes calculated by the two methods look similar. However, only the WI method has an equal probability of selection of individual terminal digit i = 0, 1, 2, …, 9. In the W method, terminal digits 1, 2, 3, and 4 have some added probability than the remaining six digits. These four terminal digits are overestimated as the probability of selection of each of these terminal digits exceeds 0.1 (see equations 7e–10e). As a result, for the W method, the sum of probabilities exceeds 1 and the sum of digit indexes exceeds 10 (see the last column of tables Appendix A1 and A2). Statistically, the sum of all ten terminal digit indexes must be '10' in any method. This is not plausible in the W method due to over selection of terminal digits 1, 2, 3, and 4. Therefore, this method fails to give a robust index value for individual terminal digits. Only the WI method is based on the mathematical formulation of the selection of equal probability for all possible terminal digits of human age (0, 1, 2, …, 9). Hence, in the digit-specified indexes, only WI methods give a robust index for all terminal digits. Summary index $W_{tot}$ based on the individual digit specified indexes with an unequal probability of selection of each terminal digit, is non-robust. In the WI method, all terminal digits have an equal probability of being included, and hence all digit-specified indexes have equal weightage for any terminal digits. Therefore, the newly proposed summary index RMWI is based on the WI method, is robust and based on a mathematical formulation. Bar charts presented in Fig 2 give the side by side comparison of digit-specific Whipple indices from Noumbissi's method ($W_i$) and the Whipple method ($WI_i$) using Turkey (2013–2018) DHS data. Clearly, terminal digit 1, 2, 3 and 4 gives overestimation in both datasets.

**An example: Turkey demographic and health survey datasets**

To strengthen and support our claim, we include the raw datasets from the Tukey demographic and health surveys (TDHS) in Appendix Table A4 for ages 21–64. We calculate new digit indexes using equations 5e to 14e with the expanded age range of 21–64 years. We named these indexes as $W_{i[21-64]}$ for i = 0, 1, 2, …, 9. Denominators of the digit indexes '1, 2, 3, and 4' ranges beyond the range 23–62 [equations: 7e, 8e, 9e, and 10e] change their digit index values. Table 4 gives three different types of digits specified indexes. The first part of Table 4 shows the digit-specific Modified Whipple index $W_i$'s (Noumbissi's method) based on the age range of 23–62 years (calculated in the previous section). The second part gives the new type of digit-specified index based on the age range of 21–64 calculated using the digit-specific Modified Whipple index $W_{i[21-64]}$ (Noumbissi's method). The third part gives the digit-specific Whipple index $WI_i$'s (calculated in the previous section). The last column of the table shows the sum of all digit indexes for all ten digits. In all six survey datasets, results for digit-specific Modified Whipple indexes showed that digit '3' is a more preferred digit than digit

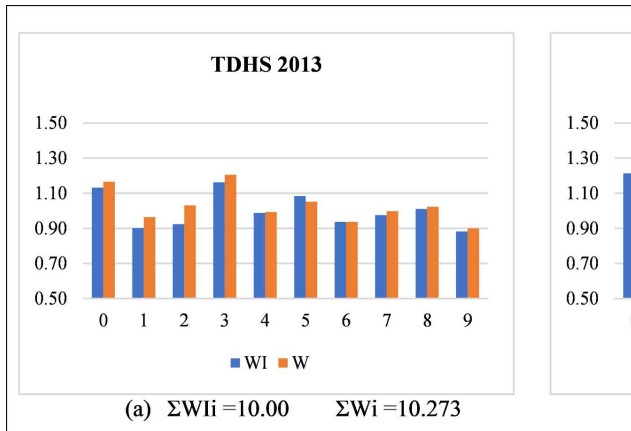
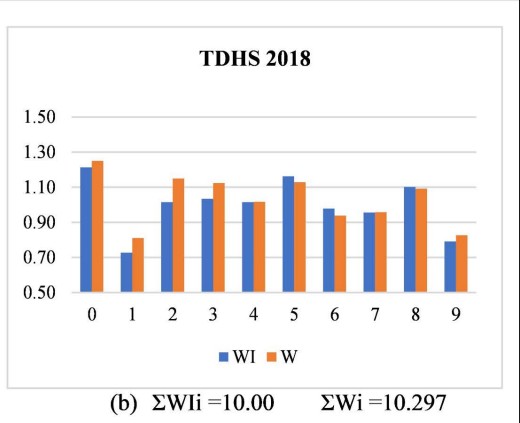

(a) ΣWIi =10.00    ΣWi =10.273        (b) ΣWIi =10.00    ΣWi =10.297

**Fig 2. Side-by-side comparison of digit-specific Whipple indices from Noumbissi's method (*Wi*) and the Whipple method (*WIi*) using Turkey (2013 and 2018) DHS data.** Data Source: Standard DHS of Turkey (2013 and 2018) (ICF, 1985–2023), Figures based on the Author's calculations of digit-specific indexes.

**Table 4. Comparison of digit-specific Whipple's indexes (*WIi*) Digit-specific modified Whipple's indexes (*Wi*), and digit-specific modified Whipple's indexes (*Wi[21–64]*) for Turkey data.**

| Survey year(s) | Digit-specific Modified Whipple index $W_i$ | | | | | | | | | | |
|---|---|---|---|---|---|---|---|---|---|---|---|
| | $W_0$ | $W_1$ | $W_2$ | $W_3$ | $W_4$ | $W_5$ | $W_6$ | $W_7$ | $W_8$ | $W_9$ | $\Sigma W_i$ |
| 1993 | 1.529 | **0.753** | **0.955** | **1.312** | **0.910** | 1.265 | 0.858 | 0.959 | 1.028 | 0.741 | **10.309** |
| 1998 | 1.306 | **0.795** | **1.042** | **1.209** | **1.028** | 1.142 | 0.950 | 0.919 | 1.079 | 0.830 | **10.299** |
| 2003 | 1.172 | **0.862** | **1.035** | **1.242** | **1.048** | 1.118 | 0.887 | 0.944 | 1.049 | 0.952 | **10.310** |
| 2008 | 1.168 | **0.823** | **1.072** | **1.161** | **1.052** | 1.135 | 0.920 | 0.932 | 1.096 | 0.913 | **10.272** |
| 2013 | 1.166 | **0.965** | **1.031** | **1.206** | **0.993** | 1.053 | 0.938 | 0.998 | 1.023 | 0.900 | **10.273** |
| 2018 | 1.251 | **0.811** | **1.150** | **1.124** | **1.017** | 1.130 | 0.938 | 0.958 | 1.092 | 0.827 | **10.297** |
| Survey year(s) | Digit-specific Modified Whipple index [Age 21–64 included] $Wi[21\_64]$ | | | | | | | | | | |
| | W0[21–64] | W1[21–64] | W2[21–64] | W3[61–64] | W4[21–64] | W5[21–64] | W6[21–64] | W7[21–64] | W8[21–64] | W9[21–64] | ΣWi[21_64] |
| 1993 | 1.529 | 0.730 | 0.907 | 1.123 | 0.837 | 1.265 | 0.858 | 0.959 | 1.028 | 0.741 | **9.977** |
| 1998 | 1.306 | 0.776 | 0.995 | 1.036 | 0.952 | 1.142 | 0.950 | 0.919 | 1.079 | 0.830 | **9.985** |
| 2003 | 1.172 | 0.841 | 0.984 | 1.057 | 0.964 | 1.118 | 0.887 | 0.944 | 1.049 | 0.952 | **9.969** |
| 2008 | 1.168 | 0.801 | 1.021 | 1.010 | 0.979 | 1.135 | 0.920 | 0.932 | 1.096 | 0.913 | **9.975** |
| 2013 | 1.166 | 0.932 | 0.969 | 1.060 | 0.932 | 1.053 | 0.938 | 0.998 | 1.023 | 0.900 | **9.971** |
| 2018 | 1.251 | 0.776 | 1.063 | 1.001 | 0.962 | 1.130 | 0.938 | 0.958 | 1.092 | 0.827 | **9.998** |
| Survey year(s) | Digit-specific Whipple index $WIi$ | | | | | | | | | | |
| | WI0 | WI1 | WI2 | WI3 | WI4 | WI5 | WI6 | WI7 | WI8 | WI9 | ΣWIi |
| 1993 | 1.407 | 0.664 | 0.813 | 1.270 | 0.937 | 1.365 | 0.883 | 0.942 | 1.018 | 0.701 | **10.000** |
| 1998 | 1.197 | 0.678 | 0.865 | 1.169 | 1.079 | 1.237 | 1.011 | 0.921 | 1.073 | 0.770 | **10.000** |
| 2003 | 1.079 | 0.754 | 0.864 | 1.236 | 1.096 | 1.207 | 0.919 | 0.939 | 1.016 | 0.891 | **10.000** |
| 2008 | 1.080 | 0.707 | 0.898 | 1.128 | 1.098 | 1.221 | 0.981 | 0.948 | 1.085 | 0.853 | **10.000** |
| 2013 | 1.132 | 0.902 | 0.925 | 1.162 | 0.989 | 1.084 | 0.937 | 0.976 | 1.010 | 0.883 | **10.000** |
| 2018 | 1.213 | 0.728 | 1.016 | 1.036 | 1.015 | 1.164 | 0.978 | 0.956 | 1.102 | 0.792 | **10.000** |

Data Source: Standard DHS datasets, Turkey (1993–2018) [18]

Authors' calculation based on the original data

'5' (see columns $W_3$ and $W_5$). However, when we use a complete range of denominators to calculate the digit-specific Modified Whipple index $W_{3[61-64]}$ for terminal digit '3', the results are reversed. Digit 5 is more preferred than digit '3' [see column $W_{3[61-64]}$ and $W_{5[61-64]}$]. This is all due to the overestimation of digit 3 in Noumbissi's method. Additionally, digit '2' is pretended as a preferred digit in five datasets, while originally the digit '2' was avoided in a few survey years. Digits 1, 2, 3, and 4 are overestimated in the digit-specific Modified Whipple index method. The pattern and rank of terminal digits calculated using the age range (21–64) are quite similar to the digit-specific Whipple index. Therefore, the digit-specific Whipple indexes give robust results as compared to the digit-specific Modified Whipple indexes $W_i$'s. The sum of all ten digits for any method must be equal to 10. The sum of digit indexes $W_i$'s are greater than 10 due to the overestimation of digits 1, 2, 3, and 4. The sum digit indexes $W_{i[21-64]}$ are very near to 10, while the sum of digit indexes $WI_i$'s are exactly equal to 10. Additionally, in raw datasets of Turkey, the sum of all ages with the terminal digit '5' is more than the terminal digit '3' (see Appendix Table A4).

### Comparison between Whipple index (*WI*), Total modified Whipple's Index (*Wtot*), and Robust Modified Whipple's Index (*RMWI*)

Comparably, all three indices give the same patterns over the years and can be used for spatial, temporal, and demographic comparisons within nations and globally. The original Whipple index is a good choice to check the quality of age data when we are interested in checking the heaping at digit 0 or 5 only. The other two indices will be used to check the overall quality of reported ages. However, digit-specific modified indexes (Noumbissi's indexes) give overestimated values for digits 1, 2, 3, and 4. This indicated a misestimation of preference for the digits 1, 2, 3, and 4. For example, Noumbissi's index for digit 3 in the Turkish dataset showed that it is more reported than digit 5. Which is not the actual case. Noumbissi's indexes calculated for digits 1, 2, 3, and 4 give the wrong representation of the original reported digit due to the inaccuracy of equations (see equations 8–10 or 8e–10e). The $W_{tot}$ based on these inconsistent equations did not provide reliable and accurate information about the quality of age data for individual digits. The new *RMWI* index is based on mathematical equations (equations 16–26) with the same methodology proposed in the original Whipple index and by Roger et al. This modification gives more reliable and robust results based on mathematical models as compared to the previous proposal by Spoorenberg.

   Although the comparison figures of $W_{tot}$ and *RMWI* both give the same pattern (Fig 3). However, $W_{tot}$, based on Noumbissi's version, is impractical for comparison and doesn't give a summary value for any dataset. It is also based on unrealistic and non-mathematical formulations and, therefore, not robust. *RMWI* is based on the Whipple index, and the whole data range is thus preferred over $W_{tot}$. *RMWI* gives more precise, robust, and accurate results as compared to $W_{tot}$.

### Simulation study

To evaluate the performance of the proposed modification, we applied it to several simulated datasets of different sizes. All series are randomly generated, and there is no observed heaping at the digits 0 and 5. Fig 4 gives the digit-specified Whipple and modified Whipple indexes for all ten digits for different series, while Fig 5 presents a side-by-side comparison of digit-specific Whipple indices from Noumbissi's method ($W_i$) and the Whipple method ($WI_i$), clearly illustrating the overestimation issue in Wi and the balanced distribution in $WI_i$. For ease of presentation, we compared *RMWI* with $W_{tot}$ and *WI*, respectively (Table 6). In 1st comparison of *RMWI* and $W_{tot}$, consider series S1, where $WI_0 = 1.229$ is most preferred followed by $WI_9$, $WI_8$, $WI_3$, $WI_2$, $WI_1$, $WI_7$, $WI_5$, $WI_6$, and $WI_4 = 0.723$ (most avoided digit) while $W_3 = 1.264$ is most preferred followed by $W_0$, $W_2$, $W_7$, $W_8$, $W_9$, $W_1$, $W_5$, $W_6$ and $W_4 = 0.843$ (most avoided). Digit 3 ranked up from place 4th to 1st, and digit 2 ranked up from place 5th to 2nd due to over-estimation of digits 2 and 3 in $W_{tot}$ or simply an over-estimation in Noumbissi's indexes. This type of over-estimation in Noumbissi's indexes makes $W_{tot}$ penniless, and $W_{tot}$ remains incapable of accessing the quality of age data. Hence, *RMWI* gives a more accurate and robust estimate for the quality of age distribution.

   In all simulated data series, the value of the Whipple index (*WI*) is at its perfection level (Table 5). However, *RMWI* says that data in Series S1 and S2 are moderate, in Series S6, fairly accurate, and in Series S7, S9, and S10 are perfectly

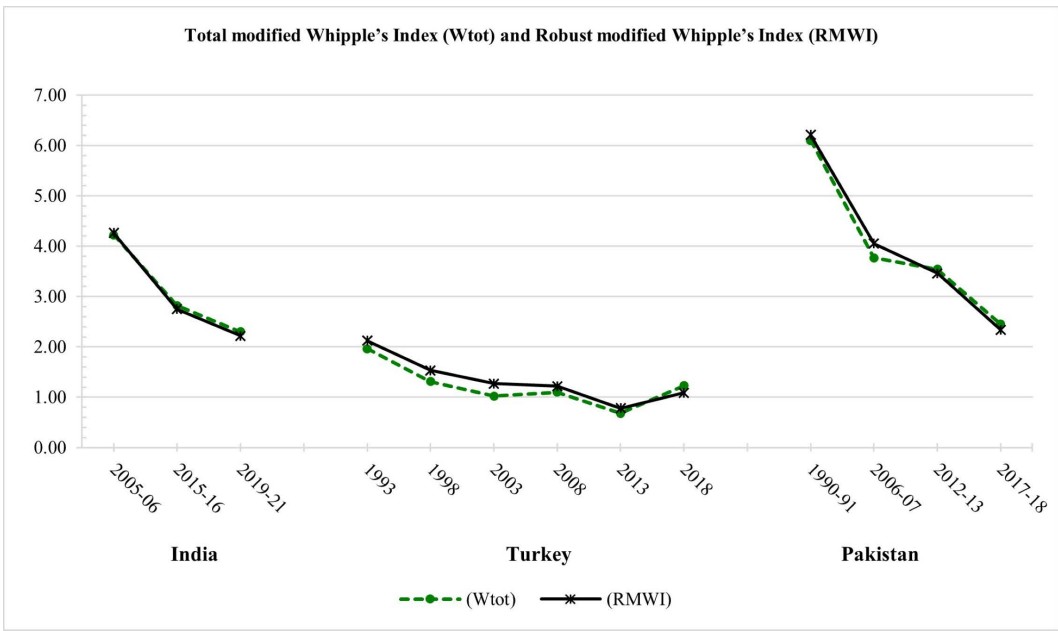

**Fig 3. Total modified Whipple's Index (*Wtot*), and Robust modified Whipple's Index (*RMWI*).**

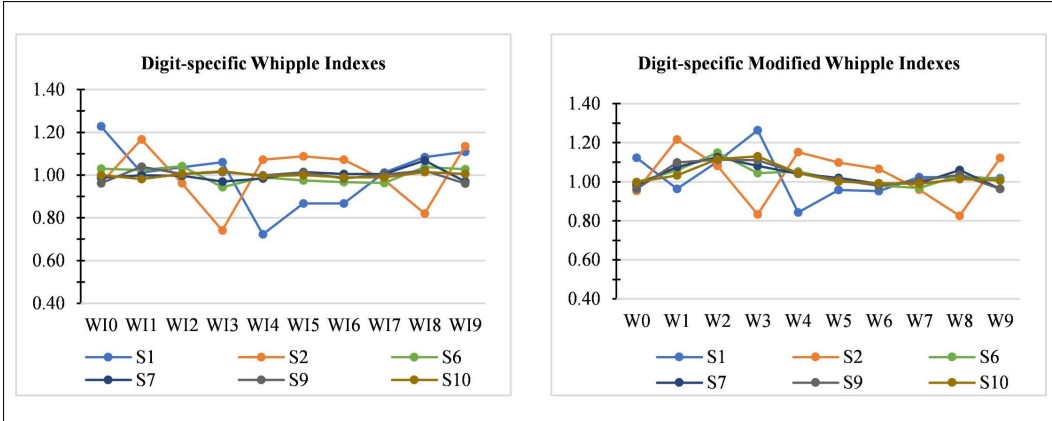

**Fig 4. Quality of age reporting: Digit-specific original Whipple's index (*WIi*) and digit-specific Modified Whipple's index (*Wi*) for the simulated data series.**

accurate. *RMWI* presents the actual picture of all data series, taking all terminal digits. We can observe that in series S1, there is a visible avoidance at digit 4 ($WI_4 = 0.723$), but *WI* ignores it at all. Similarly, in series S2, digit avoidance at digit 3 ($WI_3 = 0.741$). These preferences and avoidance are accurately covered by *RMWI* and ignored by *WI*. The *WI* emphasizes only heaping at digits 0 and 5 and ignores all other digits. Therefore, if heaping or preferences (avoidance) is present at any(some) other digit(s) (1, 2, 3, 4, 6, 7, 8, 9), *WI* fails to assess the overall quality of age distribution. Fig 5, based on simulated datasets, presents a side-by-side comparison of digit-specific Whipple indexes from Noumbissi's method ($W_i$) and the Whipple method ($WI_i$). These bar charts clearly illustrate the overestimation present in Noumbissi's method, particularly for digits 1–4, whereas the $WI_i$ used in the proposed index remain more balanced across all digits.

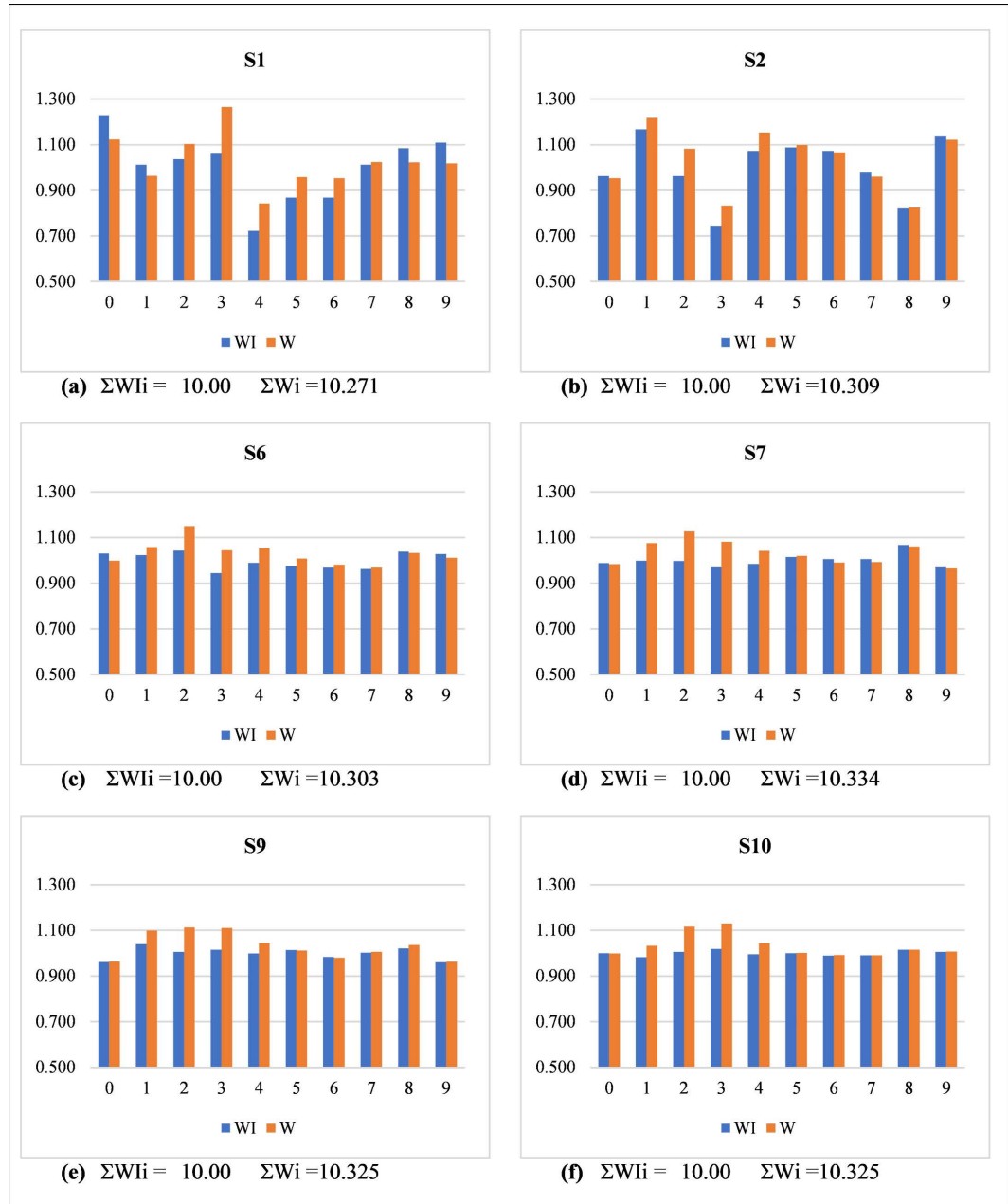

**Fig 5. Side-by-side comparison of digit-specific Whipple indices from Noumbissi's method (*Wi*) and the Whipple method (*WIi*) using Simulated datasets.**

## Discussion and conclusion

This study introduces the Robust Modified Whipple Index (*RMWI*), a new measure for assessing the quality of age data. *RMWI* addresses the limitations of earlier indices by incorporating all ten terminal digits and applying a mathematically consistent framework. The proposed *RMWI* was applied to DHS datasets from Pakistan, India and Turkey, to evaluate the misreporting of age data. A comparison with existing summary indices *WI* and $W_{tot}$ was carried out. The results revealed

**Table 5. Comparison between Whipple index (*WI*), Total modified Whipple's Index (*W<sub>tot</sub>*), and Robust modified Whipple's Index (*RMWI*).**

| Countries/ years | Original Whipple index ($WI$) | Total modified Whipple's Index ($W_{tot}$) | Robust modified Whipple's Index ($RMWI$) (Proposed) |
|---|---|---|---|
| **India** | | | |
| 2005–2006 | 204.92 | 4.22 | 4.26 |
| 2015–2016 | 165.74 | 2.82 | 2.75 |
| 2019–2021 | 150.38 | 2.30 | 2.22 |
| **Turkey** | | | |
| 1993 | 138.64 | 1.96 | 2.12 |
| 1998 | 121.69 | 1.31 | 1.53 |
| 2003 | 114.30 | 1.02 | 1.27 |
| 2008 | 115.04 | 1.10 | 1.22 |
| 2013 | 110.80 | 0.68 | 0.78 |
| 2018 | 118.84 | 1.23 | 1.09 |
| **Pakistan** | | | |
| 1990–1991 | 255.29 | 6.10 | 6.21 |
| 2006–2007 | 197.58 | 3.77 | 4.05 |
| 2012–2013 | 181.99 | 3.54 | 3.46 |
| 2017–2018 | 152.58 | 2.45 | 2.34 |

Data Source: Standard DHS of India (2005–2021), Turkey (1993–2018) and Pakistan (1990–2018) [18]

**Table 6. Comparison between Whipple index (*WI*), Total modified Whipple's Index (*W<sub>tot</sub>*), and Robust modified Whipple's Index (*RMWI*) in simulated data series.**

| Data Series (n) | Original Whipple index ($WI$) | Total modified Whipple's Index ($W_{tot}$) | Robust modified Whipple's Index (Proposed) ($RMWI$) |
|---|---|---|---|
| S1 (415) | 104.82 | 0.839 | 1.084 |
| S2 (634) | 102.52 | 1.166 | 1.073 |
| S6 (4144) | 100.27 | 0.408 | 0.323 |
| S7 (6248) | 100.11 | 0.473 | 0.185 |
| S9 (21114) | 98.78 | 0.510 | 0.192 |
| S10 (42030) | 100.00 | 0.365 | 0.086 |

that the *RMWI* consistently captures age heaping across all terminal digits and provides a more nuanced view of misreporting than existing indices. The index demonstrated a visible variation in age data quality across surveys. The 1st wave of Pakistan DHS, i.e., PDHS 1990–91, showed the highest *RMWI* value, indicating the poorest age reporting. In contrast, the PDHS 2017–18, the most recent wave, showed notable improvement, suggesting progress in data collection practices or increased respondent awareness over time. The comparative results from India and Turkey further validate the utility of the *RMWI*, as it accurately reflected expected patterns of digit preference and misreporting present in previous studies [2]. The *RMWI* results also aligned well with simulated datasets. In side by side comparison of digit indexes, Fig 5 revealed the overestimation of terminal digits $W_1$, $W_2$, $W_3$ and $W_4$, reinforcing the *RMWI* robustness and sensitivity.

In the overall summary of all three indices, the *WI* measures heaping at ages ending in digits 0 and 5, ignores distortions that occur at other terminal digits. This narrow scope underestimates the true extent of age misreporting. The $W_{tot}$ is based on Noumbissi's indexes, includes all ten digits but suffers from mathematical inconsistencies. Denominators for some digits (1, 2, 3 and 4) falling outside the selected age range lead to overestimation of these indexes and unreliable

results of the summary index. *RMWI* improves upon *Wtot* by using Rogers' digit-specific Whipple Indexes (*Wli*), each calculated with a consistent denominator (ages 23–62). This method yields more accurate digit preference and avoidance patterns and better reflects real-world and simulated data, as seen in the applications to India, Pakistan, and Turkey.

One of the key methodological decisions in constructing the *RMWI* is the use of a 10-year terminal digit structure, which assumes a uniform distribution of terminal digits from 0 to 9 in the absence of age heaping. This approach reintroduces the 10-year linearity assumption found in the original Whipple's Index and Roger's framework [14]. In contrast, Noumbissi's 5-year approach, based on a narrower linearity assumption [15] gives disproportionate weight to digits 1, 2, 3, and 4, resulting in mathematical inconsistency and unequal representation of digit preferences. By adopting a uniform 10-digit structure, the *RMWI* ensures equal weightage to all digits, allowing for a more balanced and consistent evaluation of age reporting. Although this may reduce the granularity offered by the 5-year grouping, it supports the development of a mathematically robust and generalizable summary index suitable for use across diverse populations and data sources.

To support interpretation, a new criterion classifies *RMWI* values into quality of age reporting categories: 0.00–0.19: Perfect (< 1% error), 0.20–0.99: Fairly accurate, 1.00–1.99: Moderate, 2.00–2.99: Poor and ≥3.00: Very poor (≥ 15% error). These thresholds correspond to estimated levels of age misreporting and offer a practical tool for comparing datasets. The inclusion of an approximate percentage of people reporting incorrect age associated with each quality category further enhances the practical utility of the criterion.

Hence, by taking the digit-specific indexes for all ten terminal digits based on Regar's Whipple method, the *RMWI* provides a more accurate summary measure of the overall quality of age reporting compared to both *WI* and $W_{tot}$. The two main advantages of the proposed *RMWI* over *WI* and $W_{tot}$ make it more useful in various aspects.

1. The original *WI* is limited to capture heaping at a multiple of five (digits 0 and 5), whereas *RMWI* covers all terminal digits.

2. *RMWI* offers a more robust and mathematically consistent approach than $W_{tot}$.

Therefore, to assess the overall quality of age distribution, *RMWI* is a better choice than *WI* and $W_{tot}$. The proposed index is fully applicable for spatial, temporal, and sociodemographic comparisons in assessing the quality of age data. Its standardized calculation and interpretive criterion allow for meaningful comparisons of age data quality across different geographic regions, over time, and among various population subgroups. In this study, the *RMWI* was applied to datasets from two South Asian countries and Turkey. Future research should test its applicability across a broader geographic scope, including sub-Saharan Africa, to further demonstrate its robustness and generalizability across diverse demographic settings. Quality of age reporting can also be compared by other demographic characteristics such as gender, education, or socioeconomic status. Finally, the proposed index is simple to calculate, fairly constructed on mathematical formulation, and fully based on all age digits. It is practically compatible with the original Whipple's index and the total modified Whipple index and provides a more robust measure of the overall quality of age reporting.

The proposed index is not only applicable to age data collected during interviews, surveys or censuses, but it is equally applicable for assessing the digit preference and quality of age reporting in human mortality data. Ongoing demographic and health transitions in the Global South call for better and more comprehensive demographic data covering the entire age span, including working and old age. Therefore, precise identification of age-reporting problems is crucial for producing reliable population exposures and demographic rates in the region. Further research is needed to develop plausible and transparent adjustment methods to address distortions in survey or census based measures.

## Supporting information

**S1 Appendix.**
(DOCX)

## Acknowledgments

Afza Rasul is grateful to the Higher Education Commission of Pakistan for awarding a stipend to visit the Max Planck Institute for Demographic Research (MPIDR) for Ph.D. Research under the Faculty Development Program for Pakistani Universities and to the MPIDR for welcoming as a guest researcher and providing a research platform.

## Author contributions

**Conceptualization:** Afza Rasul, Jamal Abdul Nasir.

**Data curation:** Afza Rasul.

**Formal analysis:** Afza Rasul, Domantas Jasilionis.

**Investigation:** Afza Rasul, Jamal Abdul Nasir.

**Methodology:** Afza Rasul, Jamal Abdul Nasir, Dmitri A. Jdanov.

**Software:** Afza Rasul.

**Supervision:** Jamal Abdul Nasir, Dmitri A. Jdanov.

**Writing – original draft:** Afza Rasul, Domantas Jasilionis, Dmitri A. Jdanov.

**Writing – review & editing:** Afza Rasul.

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
