## [Decision Letter · Decision Letter 0]

6 Jul 2025

PONE-D-25-11698Quality of reported ages: A robust re-modification in Total Modified Whipple’s IndexPLOS ONE

Dear Dr. Rasul,

Thank you for submitting your manuscript to PLOS ONE. After careful consideration, we feel that it has merit but does not fully meet PLOS ONE’s publication criteria as it currently stands. Therefore, we invite you to submit a revised version of the manuscript that addresses the points raised during the review process.

We look forward to receiving your revised manuscript.

Kind regards,

Abroon Qazi, Ph.D.

Academic Editor

PLOS ONE

Journal Requirements:

2. In the online submission form, you indicated that [The datasets used in this research are available in the Demographic and Health Surveys (DHS) Program repository www.dhsprogram.com. Demographic and Health survey data sets are freely available on request. Prior permission has been taken before using the data set from DHS.].

Additional Editor Comments:

Please incorporate more recent publications that are relevant to the theme of the study. The manuscript lacks a strong theoretical foundation in its current form.

Reviewers' comments:

Reviewer's Responses to Questions

**Comments to the Author**

1. Is the manuscript technically sound, and do the data support the conclusions?

Reviewer #1: Yes

Reviewer #2: Yes

2. Has the statistical analysis been performed appropriately and rigorously? 

Reviewer #1: Yes

Reviewer #2: Yes

3. Have the authors made all data underlying the findings in their manuscript fully available?

Reviewer #1: Yes

Reviewer #2: Yes

4. Is the manuscript presented in an intelligible fashion and written in standard English?

Reviewer #1: Yes

Reviewer #2: Yes

5. Review Comments to the Author

Reviewer #1: To be accepted for publication in PLOS One, research articles must satisfy the following criteria:

1. The study presents the results of original research.

Yes. The manuscript presents an original methodological contribution to demographic measurement by proposing a new index, “RWtot”, for assessing the quality of age reporting. The innovation lies in its mathematically robust formulation and its ability to overcome limitations in existing indices like the Total Modified Whipple Index (Wtot) and Noumbissi’s digit-specific indices. The proposed method enhances accuracy and interpretability for assessing age heaping across all digits.

2. Results reported have not been published elsewhere.

The approach and formulation of the RWtot index appear to be novel and are applied to publicly available Demographic and Health Survey (DHS) datasets and simulated data for validation.

3. Experiments, statistics, and other analyses are performed to a high technical standard and are described in sufficient detail.

Yes. The authors provide clear mathematical derivations, define all index formulas, and apply the method to extensive real-world datasets from India, Pakistan, and Turkey. They also include simulation studies that support the index's robustness. The statistical procedures and underlying assumptions (e.g., equal terminal digit probabilities) are well justified.

4. Conclusions are presented in an appropriate fashion and are supported by the data.

Yes. The conclusions follow logically from the empirical and simulated results. The authors demonstrate that RWtot produces more balanced and accurate assessments of digit preferences. Figures and tables effectively support these claims.

5. The article is presented in an intelligible fashion and is written in standard English.

Generally yes. The article is intelligible and the structure is coherent. However, minor language improvements are recommended.

6. The research meets all applicable standards for the ethics of experimentation and research integrity.

Yes. The research uses publicly available DHS data. No human or animal subjects are directly involved.

7. The article adheres to appropriate reporting guidelines and community standards for data availability.

Yes. Data used in the study are available through the DHS Program website.

Summary of the Article

This paper introduces a more precise measure for detecting age heaping in demographic data by proposing a re-modified version of the Total Modified Whipple’s Index (RWtot). The authors critique the limitations of existing age-quality measures—particularly the Whipple Index and Spoorenberg's Total Modified Whipple Index—arguing that these either focus too narrowly on digits ending in 0 and 5 or suffer from biased digit-specific weighting. The proposed RWtot, grounded in the original Whipple’s structure, ensures equal treatment of all terminal digits (0–9) and addresses methodological inconsistencies in earlier adaptations. The method is tested using DHS data from India, Pakistan, and Turkey and validated with simulated datasets.

General Assessment

This is an excellent and well-structured contribution to the field of demographic data quality. The authors succeed in both identifying a nuanced problem—bias in existing digit-specific indices—and offering a mathematically sound and interpretable solution. The paper is methodologically rigorous, clearly written, and practically relevant for demographers and statistical agencies working with census or survey data. Its novelty lies in the construction of RWtot using consistent probabilistic and mathematical logic, while retaining computational simplicity.

3. Major Comments

Main Strengths:

The article makes a clear and justified methodological advancement by proposing RWtot. The RWtot is built on a solid critique of the denominator issues in Noumbissi's Wi index and addresses them using a uniform framework (equations 15–24). The authors have provided convincing empirical tests using large DHS datasets and simulation studies. The work is timely and relevant for low- and middle-income countries, where age misreporting is common and demographic indicators depend on clean data.

Suggestions for Strengthening the Paper:

Robustness Checks: Consider including additional countries (e.g., sub-Saharan Africa) to show geographic robustness.

Sensitivity to Age Range: While the focus is rightly on 23–62, a brief note on the applicability (or adjustment needs) for other age ranges could broaden utility.

Terminology Consistency: Throughout the article, ensure consistent use of symbols (e.g., WI, Wi, RWtot) to avoid reader confusion, especially in equations and figures.

4. Minor Comments

Ensure that all figures and tables are referenced in the text and are clearly labeled in the final version.

5. Recommendation

Accepted with Minor Revisions

The manuscript makes a meaningful and methodologically robust contribution to demographic measurement. The paper is accepted with minor revision, especially focus the stylistic and presentation will further improve the clarity and accessibility to the readers.

Reviewer #2: Review of Manuscript

Quality of reported ages: A robust re-modification in Total Modified Whipple's Index

Summary and Overall Impression

I would like to commend the authors for this well-researched and timely manuscript. The paper addresses a fundamental and persistent challenge in demographic data analysis the accurate measurement of age misreporting. The work makes a significant and valuable contribution by proposing a new, robust summary index (RWtot) for measuring age heaping across all ten terminal digits.

The importance of this paper lies not only in the introduction of a new method but also in its rigorous and clear critique of the mathematical shortcomings in existing comprehensive indices. Specifically, the authors’ deconstruction of the Total Modified Whipple Index (Wtot) , as proposed by Spoorenberg(2007) based on Noumbissi’s work, is a critical insight for the field. This manuscript is a fine example of methodological refinement that promises to improve the quality and reliability of demographic analysis.

A major strength of the manuscript is its clear and logical deconstruction of the evolution of Whipple’s Index, from its original form to its various modifications. The critique of Noumbissi’s (1992) modification and the subsequent Spoorenberg index(Wtot) is particularly compelling. The authors effectively demonstrate the mathematical flaw wherein the formulas for terminal digits 1,2,3 and 4 lead to an overestimation of preference.

The authors employ a highly effective multi-pronged validation approach that substantially strengthens their claims.

Suggestions for Improvement

The manuscript is strong, and the following suggestions are offered in a constructive spirit to enhance its clarity, impact, and utility for the research community.

Refine the Terminology

While the name “remodified of total modified Whipple’s Index” (and its abbreviation RWtot) is accurate in describing its lineage, it is somewhat cumbersome and may hinder recall and adoption. To facilitate easier reference and citation, the authors might consider a more concise and memorable name. Suggestions include

• Comprehensive Rogers-Whipple Index (CRWI), to credit the methodological foundation.

• Ten-Digit Whipple Index (TDWI) , to highlight its key feature.

• Robust General Whipple Index (RGWI) , to emphasize its stability.

A more distinct name could help establish the index as a new standard in literature.

Address the Linearity Assumption Trade-off

The paper correctly notes that Noumbissi’s method was developed partly to address the “crude” 10-year linearity assumption of Roger’s index , opting instead for a more granular 5-year assumption. The proposed RWtot, by building on Roger’s framework, reverts to this 10-year assumption. The manuscript would be strengthened by a brief discussion explicitly acknowledging this trade-off. The authors should clarify why accepting the broader 10-year linearity assumption is a worthwhile compromise to achieve a mathematically consistent and robust summary index that covers all ten digits without distortion.

Elaborate on the Proposed Quality Criteria

The paper proposes a new set of quality criteria for the RWtot index in Table 2. This is an excellent and necessary step for practical application. However, the derivation of these specific thresholds (e.g. ,“Moderate” for a value of 1.00-1.99, corresponding to 5-9.99% of people reporting an incorrect age) is not detailed. The manuscript would be significantly enhanced by adding a paragraph explaining the rationale behind these ranges. How were these thresholds calibrated? A clearer justification would increase transparency and boost the confidence of researchers looking to apply these standards in their own work.

Enhance Data Visualization

The figures are informative, but the comparison between the competing indices could be made more direct and impactful. I would suggest creating a single, combined chart for a key dataset that plots the digit preference indices side-by-side. For example , using the Turkey 1993 data from Table 4 , a bar chart could directly contrast Noumbissi’s W; with the proposed WI; for each digit. This would provide a powerful , at-a-glance visualization of the overestimation issue in the former method and the more plausible distribution of the latter.

Concluding Remarks and Recommendation

In conclusion , this manuscript presents a valuable, robust, and well-validated tool for demographic analysis. It successfully identifies and rectifies a significant mathematical flaw in a widely used method for assessing age data quality. The proposed RWtot index is a clear improvement, offering researchers a more reliable and comprehensive measure of age misreporting.

The paper’s strengths are substantial , and the identified areas for improvement are addressable through minor revisions that would further clarify its contributions and enhance its practical utility.

Recommendation: I recommend this manuscript for publication. The proposed index represents a significant and needed improvement over existing methods for assessing the overall quality of age reporting. I am confident that this work will be of great interest and utility to the demographic research community. Publication is recommended following minor revisions to address the constructive suggestions outlined in this review.

6. PLOS authors have the option to publish the peer review history of their article (what does this mean? ). If published, this will include your full peer review and any attached files.

**Do you want your identity to be public for this peer review?** For information about this choice, including consent withdrawal, please see our Privacy Policy .

Reviewer #1: No

Reviewer #2: No

---

## [Author Response · Author response to Decision Letter 1]

29 Jul 2025

Research Article

Quality of reported ages: A robust re-modification in Total Modified Whipple’s Index

Dear Editor,

We are grateful for the opportunity to revise the manuscript titled “Quality of reported ages: A robust re-modification in Total Modified Whipple’s Index” (PONE-D-25-11698). We thank the academic editor and the reviewers for their constructive and detailed comments on the original manuscript. We have carefully considered every suggestion and have revised the manuscript accordingly. Please find below our detailed response to the editor’s and reviewers’ comments.

Note: Editor’s and reviewers’ comments are left-aligned and italicized; authors’ responses are justified and in normal text.

Response to Editor

We thank the editor for his constructive and detailed comments on the original version of the manuscript. Below, we provide a point-by-point response to each comment.

We thank the academic editor for this important observation. We carefully reviewed and updated the reference list to ensure that all entries are complete, accurate, and properly formatted. To the best of our knowledge, there is NO retracted article cited in text as well as in the reference list.

# Additional Editor Comments:

Please incorporate more recent publications that are relevant to the theme of the study. The manuscript lacks a strong theoretical foundation in its current form.

We sincerely thank the academic editor for this important feedback. We have addressed the comment in two ways:

Recent Literature: We have reviewed and incorporated several recent publications that are directly relevant to age misreporting and age data quality. These references have been added to the Introduction section (Paragraphs 1 and 2) to strengthen the manuscript’s connection with current research.

Theoretical Foundation: To enhance the theoretical basis of our study, we have added a paragraph on the theoretical foundation that clarifies how our proposed Robust General Whipple Index (RGWI) builds upon to address the limitations of earlier indices. This has been added in the revised version of the manuscript under the Introduction section, subsection: Theoretical Foundation.

Response to Reviewer # 1

We sincerely thank Reviewer 1 for the positive and encouraging feedback on our manuscript. We also appreciate the thoughtful suggestions, which have helped us to improve the manuscript further. Below, we respond point by point to each suggestion.

Suggestions for Strengthening the Paper:

Robustness Checks: Consider including additional countries (e.g., sub-Saharan Africa) to show geographic robustness.

Thank you for this helpful suggestion. As the primary focus of this study is on proposing a new methodological index, including additional regions/countries, is beyond the scope of this article. However, we agree that applying the index to other regions (such as sub-Saharan Africa) could further demonstrate its utility. We have added a note in the discussion section to suggest this as a direction for future research (Section: Discussion and Conclusion, line # 531-536). The proposed index is designed to be broadly applicable to census and survey data across diverse settings.

Sensitivity to Age Range: While the focus is rightly on 23–62, a brief note on the applicability (or adjustment needs) for other age ranges could broaden utility.

We thank the reviewer for this thoughtful comment. We already mentioned in the original manuscript (Section: Background, lines # 60-61), and now in the revised manuscript (Section: Introduction, lines # 73-75) that ‘Younger ages (0–22) and older ages (63 and above) are excluded as the linearity assumption is not plausible for these ages. However, we have also added a paragraph in the revised manuscript (Section: Methods, lines #224-231) to explain our choice of the 23–62 age range, which is commonly used in demographic studies to minimize the effects of age misreporting among children and the elderly. We also note that the proposed index can be adapted to other adult age ranges depending on data quality and the study context. However, age heaping is typically most pronounced within the selected range, making it the most appropriate for demonstrating the index’s effectiveness.

Terminology Consistency: Throughout the article, ensure consistent use of symbols (e.g., WI, Wi, RWtot) to avoid reader confusion, especially in equations and figures.

Thank you for noting this. We have carefully reviewed the entire manuscript to ensure consistent use of abbreviations and symbols. RWtot is now referred to as RMWI (Robust Modified Whipple Index) throughout the revised text. The original Whipple Index is abbreviated as WI; Noumbissi’s digit-specific indices are denoted as Wi, Spoorenberg’s index is denoted as Wtot, Rogers’ Whipple digit-specific indices are denoted as WIi.

Minor Comments: Ensure that all figures and tables are referenced in the text and are clearly labeled in the final version.

We have reviewed the manuscript and ensured that all figures and tables are correctly numbered, labeled, and referenced in the body of the text.

Response to Reviewer # 2

We sincerely thank the reviewer for the recognition of the manuscript’s contributions and appreciate the constructive feedback provided to enhance its clarity, impact, and utility. Below, we respond point by point to each suggestion.

Suggestions for Improvement

The manuscript is strong, and the following suggestions are offered in a constructive spirit to enhance its clarity, impact, and utility for the research community.

Refine the Terminology

While the name “remodified of total modified Whipple’s Index” (and its abbreviation RWtot) is accurate in describing its lineage, it is somewhat cumbersome and may hinder recall and adoption. To facilitate easier reference and citation, the authors might consider a more concise and memorable name. Suggestions include:

• Comprehensive Rogers-Whipple Index (CRWI), to credit the methodological foundation.

• Ten-Digit Whipple Index (TDWI), to highlight its key feature.

• Robust General Whipple Index (RGWI), to emphasize its stability.

A more distinct name could help establish the index as a new standard in literature.

We thank the reviewer for this thoughtful observation and valuable suggestion. We agree that a concise and memorable name could enhance the visibility and adoption of the proposed index. After careful consideration, we have adopted the term “Robust Modified Whipple Index (RMWI)”. RMWI is a more concise, memorable, and meaningful name that aligns with the manuscript title. This revised terminology emphasises both the methodological robustness of the index and its origin in Whipple’s modification tradition. The abbreviation RMWI has been used consistently throughout the revised manuscript.

Address the Linearity Assumption Trade-off

The paper correctly notes that Noumbissi’s method was developed partly to address the “crude” 10-year linearity assumption of Roger’s index , opting instead for a more granular 5-year assumption. The proposed RWtot, by building on Roger’s framework, reverts to this 10-year assumption. The manuscript would be strengthened by a brief discussion explicitly acknowledging this trade-off. The authors should clarify why accepting the broader 10-year linearity assumption is a worthwhile compromise to achieve a mathematically consistent and robust summary index that covers all ten digits without distortion.

We thank the reviewer for this insightful observation. We acknowledge that returning to the 10-year linearity assumption represents a methodological trade-off. However, the primary goal of the proposed Robust Modified Whipple Index (RMWI) is to ensure mathematical consistency and equal weightage across all ten terminal digits (0–9). This uniform structure avoids the imbalance and overestimation seen in digit-specific indexes calculated by using Noumbissi’s method due to varying denominators in digits 1, 2, 3, and 4 (see Introduction subsection: Mathematical inconsistencies in Noumbissi’s Indexes). By adopting a 10-digit framework, the RMWI provides a more balanced and interpretable summary of age heaping patterns across the full digit spectrum. We believe this trade-off supports our goal of developing a robust and generalizable index for assessing age misreporting. In the revised manuscript, we have added a paragraph in the Discussion and Conclusion section (Paragraph 3, lines # 503-513) to explain the rationale behind this choice.

“One of the key methodological decisions in constructing the Robust Modified Whipple Index (RMWI) ………………………………….……………. mathematically robust and generalizable summary index suitable for use across diverse populations and data sources.”

Elaborate on the Proposed Quality Criteria

The paper proposes a new set of quality criteria for the RWtot index in Table 2. This is an excellent and necessary step for practical application. However, the derivation of these specific thresholds (e.g. ,“Moderate” for a value of 1.00-1.99, corresponding to 5-9.99% of people reporting an incorrect age) is not detailed. The manuscript would be significantly enhanced by adding a paragraph explaining the rationale behind these ranges. How were these thresholds calibrated? A clearer justification would increase transparency and boost the confidence of researchers looking to apply these standards in their own work.

The thresholds presented in Table 2 are derived from interpreting the RMWI value as a deviation from ideal terminal digit index values. A value of 0.00 reflects perfect uniformity in terminal digit reporting, indicating no age heaping and misreporting, while higher values represent increasing levels of digit preference or avoidance. A deviation of less than 1% is considered perfectly accurate. Values between 1% and 4.99% are classified as fairly accurate, while the critical threshold falls between 5% and 9.99%, where clear signs of age misreporting are evident, named as moderate, showing approximate data. These cutoffs draw a conceptual parallel with the conventional thresholds used in statistical hypothesis testing. Just as p-values below 0.01, 0.05, and 0.10 represent varying levels of statistical significance in social science research, the RMWI categories provide meaningful breakpoints to gauge the severity of age misreporting. For example, <1% deviation aligns with a highly accurate dataset (analogous to a highly significant p-value), 5–10% marks the zone where misreporting becomes substantively important (like a significant finding), and deviations above 10% indicate severe inaccuracies in age data, similar to a non-significant or unreliable result in statistical terms. We added a Paragraph in the Method section to explain these thresholds immediately after Table 2 (lines # 258-273).

Enhance Data Visualization

The figures are informative, but the comparison between the competing indices could be made more direct and impactful. I would suggest creating a single, combined chart for a key dataset that plots the digit preference indices side-by-side. For example, using the Turkey 1993 data from Table 4 , a bar chart could directly contrast Noumbissi’s W; with the proposed WI; for each digit. This would provide a powerful , at-a-glance visualization of the overestimation issue in the former method and the more plausible distribution of the latter.

Thank you. We fully agree with the reviewer’s suggestion. To enhance the clarity and visual impact of the comparison between indices, we have added a combined bar chart using the Turkey datasets (Section: Comparison of digit-specific indexes: Wi vs WI, Fig 2) and the simulated dataset (Section: Fig 5). These bar charts give a direct, side-by-side comparison of Noumbissi’s digit-specific indices (Wi) and Rogers’s Whipple’s digit indices (WIi), clearly illustrating the overestimation issue in the Wi and the balanced distribution in WIi.

---

## [Editor Report · Decision Letter 1]

4 Aug 2025

Quality of reported ages: A robust re-modification in Total Modified Whipple’s Index

PONE-D-25-11698R1

Dear Ms Rasul,

We’re pleased to inform you that your manuscript has been judged scientifically suitable for publication and will be formally accepted for publication once it meets all outstanding technical requirements.

Kind regards,

Abroon Qazi, Ph.D.

Academic Editor

PLOS ONE

Additional Editor Comments (optional):

The authors have adequately addressed all reviewer comments.
---

## [Editor Report · Acceptance letter]

PONE-D-25-11698R1

PLOS ONE

Dear Dr. Rasul,

I'm pleased to inform you that your manuscript has been deemed suitable for publication in PLOS ONE. Congratulations! Your manuscript is now being handed over to our production team.

Kind regards,

on behalf of

Dr. PLOS Manuscript Reassignment

Staff Editor

PLOS ONE